# Biomolecules to Biomarkers? U87MG Marker Evaluation on the Path towards Glioblastoma Multiforme Pathogenesis

**DOI:** 10.3390/pharmaceutics16010123

**Published:** 2024-01-18

**Authors:** Markéta Pokorná, Viera Kútna, Saak V. Ovsepian, Radoslav Matěj, Marie Černá, Valerie Bríd O’Leary

**Affiliations:** 1Department of Medical Genetics, Third Faculty of Medicine, Charles University, Ruská 87, Vinohrady, 10000 Prague, Czech Republic; marketa.pokorna@lf3.cuni.cz (M.P.); marie.cerna@lf3.cuni.cz (M.Č.); 2Department of Experimental Neurobiology, National Institute of Mental Health, Topolová 748, 25067 Klecany, Czech Republic; viera.kutna@nudz.cz; 3Faculty of Engineering and Science, University of Greenwich London, Chatham Maritime, Kent ME4 4TB, UK; s.v.ovsepian@greenwich.ac.uk; 4Department of Pathology, Third Faculty of Medicine, Charles University, Ruská 87, Vinohrady, 10000 Prague, Czech Republic; radoslav.matej@ftn.cz; 5Department of Pathology, University Hospital Královské Vinohrady, Šrobárova 50, Vinohrady, 10000 Prague, Czech Republic

**Keywords:** prominin-1, ICAM-1, lncRNA, Glioma, Eker rats, U87MG, *PARTICLE*, *GAS5*, GBM, Biomarker

## Abstract

The heterogeneity of the glioma subtype glioblastoma multiforme (GBM) challenges effective neuropathological treatment. The reliance on in vitro studies and xenografted animal models to simulate human GBM has proven ineffective. Currently, a dearth of knowledge exists regarding the applicability of cell line biomolecules to the realm of GBM pathogenesis. Our study’s objectives were to address this preclinical issue and assess prominin-1, ICAM-1, *PARTICLE* and *GAS5* as potential GBM diagnostic targets. The methodologies included haemoxylin and eosin staining, immunofluorescence, in situ hybridization and quantitative PCR. The findings identified that morphology correlates with malignancy in GBM patient pathology. Immunofluorescence confocal microscopy revealed prominin-1 in pseudo-palisades adjacent to necrotic foci in both animal and human GBM. Evidence is presented for an ICAM-1 association with degenerating vasculature. Significantly elevated nuclear *PARTICLE* expression from in situ hybridization and quantitative PCR reflected its role as a tumor activator. *GAS5* identified within necrotic GBM validated this potential prognostic biomolecule with extended survival. Here we present evidence for the stem cell marker prominin-1 and the chemotherapeutic target ICAM-1 in a glioma animal model and GBM pathology sections from patients that elicited alternative responses to adjuvant chemotherapy. This foremost study introduces the long non-coding RNA *PARTICLE* into the context of human GBM pathogenesis while substantiating the role of *GAS5* as a tumor suppressor. The validation of GBM biomarkers from cellular models contributes to the advancement towards superior detection, therapeutic responders and the ultimate attainment of promising prognoses for this currently incurable brain cancer.

## 1. Introduction

Specialized glial cells, known as astrocytes, outnumber neurons in the central nervous system (CNS) by over fivefold [1,2]. Furthermore, astrocytes support the detoxification of blood capillary networks and guidance during migration [3,4]. Deregulation of astrocytic pathways along with the build-up of somatic mutations in glial progenitor cells empower malignant glioma formation, a subset of which is glioblastoma multiforme (GBM) [5,6]. Despite its relatively low occurrence rate in comparison to all cancer types, GBM (WHO grade IV) is an aggressive high-grade oligo—astrocytoma with mixed cellular features and, on average, a maximum fifteen-month median survival rate [7,8]. The heterogeneous morphological features of GBM make classification a challenge for neuropathologists due to the presence of diverse tumor and non-tumor cell types along with microenvironmental components [9,10]. Notwithstanding advances in conventional treatments involving surgical resection followed by radiotherapy and adjuvant chemotherapy interventions as well as electric tumor-treating fields (TTFields) [11], GBM remains incurable due to therapeutic resistance and its structural infiltrative characteristics [12,13]. The incorporation of molecular data, such as Ki-67, isocitrate dehydrogenase (IDH) and *BRAF* V600E mutations, have been recognized as potential prognostic and predictive biomarkers of GBM [14,15]. Recently, the fifth edition of the WHO Classification of Tumors of the Central Nervous System has greatly emphasized the importance of molecular parameters to grade gliomas and estimate prognosis more precisely [16]. Current genetic markers of glioma also include codeletions of chromosomal arms 1p and 19q, histone H3F3A alterations, nuclear alpha-thalassemia/mental retardation X-linked syndrome (ATRX) gene mutations, O6-methylguanine-DNA methyltransferase (MGMT) promoter methylation status, loss of cyclin-dependent kinase inhibitor 2A (CDKN2A), epidermal growth factor receptor (EGFR) amplification, combined gain of chromosome 7 and loss of chromosome 10 and telomerase reverse transcriptase (TERT) promoter mutations [17]. Further research is required to fully assess their prognostic value in the context of glioblastoma and their influence on treatment selection. Immunotherapy is at an early stage with tyrosine kinase inhibitors (e.g., gefitinib and erlotinib), proving unsuccessful in GBM phase II clinical trials [18]. New GBM molecular markers are required from recently recognized biological avenues, such as tissue-specific long non-coding RNAs and their epigenetic influence [17]. Advanced statistical methodology are warranted to assist in prognostic accuracy [19]. The current issue is that an imperfect understanding of GBM potentially accounts for such failures and has hastened the need to identify nuances in associated biocompatible nano-molecules to combat the challenge in visualization and grading GBM tumors towards durable effective therapeutic responses [20,21]. Microscopic features of GBM include microvascular proliferation, hypercellularity, high mitotic count, nuclei atypica and pseudo-palisades adjacent to hypoxic and necrotic foci [22,23]. Simulating human GBM has relied heavily on in vitro cell line platforms and xenografted animal models, yet the data from such sources has generally proven untranslatable in clinical trials focused on treatment pathways [24,25].

Recently, we validated and deciphered the phenotypic nuances of the GBM cell line U87MG (ECACC 89081402; p53 wild-type; methylated MGMT promoter) [2,26,27]. Findings revealed that exposure to all-trans retinoic acid (ATRA) attenuated the expression of prominin-1 (CD133), a stem cell marker. Histochemical evidence highlighted the presence of prominin-1 in the cytoplasm but its absence in perinuclear space [26]. Conversely, the intercellular adhesion molecule-1 (ICAM-1; CD54), a chemotherapeutic targeting molecule, showed increased expression in response to ATRA [26]. ICAM-1 is regarded as an important mediator in tumor migration and invasion [28,29]. The long non-coding RNA (lncRNA) *PARTICLE*, a negative regulator of tumor suppressors, demonstrated significantly high expressivity in early passaged U87MG exposed to ATRA with transcripts confined to the leading edge of the microenvironment [26]. Nevertheless, low transcript levels of the lncRNA *GAS5* were noted in this location under similar conditions [26], reflective of its tumor suppressor status and association with overall GBM patient survival [30].

The objective of this study was to evaluate previously obtained in vitro U87MG derived data within rodent and human ex vivo GBM platforms. A germline insertion of the tuberous sclerosis (*Tsc2*) gene gave rise to the established Eker rat model [31]. Our previous investigations in aged Tsc2 +/− rats determined hallmarks of gliomas [32]. This study focused on the Eker rat model and four GBM patients representing responders or non-responders to temozolomide chemotherapy. Evidence of prominin-1 and ICAM-1 is presented and their pertinence toward GBM patient histopathology is considered. This foremost examination highlights the expression profile of lncRNA *PARTICLE* in GBM derived from patients and validates *GAS5* in this context, providing insights into potentially valuable targeting biomolecules for GBM therapeutics.

## 2. Materials and Methods

This research focused on the examination of histological sections obtained from patients displaying the classical pattern of GBM and elicited various responses to chemotherapy. Where possible, analysis was performed in parallel with sections from an Eker rat glioma model (inclusion criteria: 18-month-old males, subcortical tumors in the cerebrum, tsc2 +/− genotype [32,33]). All experimental procedures were approved by the local Committee for Animal Protection and were conducted in accordance with the Animal Protection Code of the Czech Republic and the directive of the European Community Council on the use of laboratory animals (2010/63/EC).

### 2.1. Patient Diagnosis and Pathology

Patients diagnosed with GBM (Table 1), who underwent surgical resection with/without temozolomide chemotherapy, were part of a standard therapeutic intervention at the University Hospital Královské, Vinohrady, Prague, Czech Republic. Tissue sections (5 μm) were cut with a microtome from paraffin embedded GBM and stained for morphological and molecular biomarker readouts as indicated below.

### 2.2. Haematoxylin and Eosin Staining of Human GBM Pathogenesis

Slides were deparaffinized in xylene (3 × 10 min), submerged in absolute ethanol (3 × 5 min) and rehydrated in a 95–70% ethanol series (5 min). Slides were washed in running water (2 min) and then stained in Mayer’s haematoxylin solution (3 min) according to standard protocols [34]. After washing in running water (15 min), slides were immersed in eosin solution (2 min) and subsequently rinsed in 70% ethanol and 95% ethanol. Slides were dehydrated in ethanol (100%, 5 × 3 min), cleared in xylene (10 × 3 min), cover-slipped and viewed under an Olympus BX43F light microscope using the 10× and 40× objectives.

### 2.3. Immunohistochemistry for Prominin-1, ICAM-1 and/or GFAP Astrocyte Marker in Cortical Sections

Following deparaffinization and fixation in absolute ethanol (outlined above), pathological tumor sections were permeabilized in PBST (PBS, 0.4% Triton X-100) for 30 min and washed in PBS (10 × 3 min). Sections were exposed to a blocking solution (5% bovine serum albumin in PBS). The presence of prominin-1 or ICAM-1 was determined with incubation (O/N 4 °C; 1: 200 dilution in blocking solution) in rabbit monoclonal anti-prominin-1/CD133 (cat. # ab216323; Abcam, Cambridge, UK) or rabbit monoclonal anti-ICAM-1/CD54 (cat. # 4915, Cell Signaling Technology, Danvers, MA, USA), respectively. Following washes in PBST (15 × 3 min), sections were incubated in the dark in Alexa fluor^®^488 goat anti-rabbit IgG (H + L) (cat. # A-11008; Thermo Scientific (Waltham, MA USA); 1:500; 1 h RT in blocking solution). The protocol continued as required for double immunostaining with the astrocyte marker glial fibrillary acidic protein (GFAP). After washing in PBS (15 min × 3), the presence of GFAP was determined with incubation (O/N 4 °C; 1:200 dilution in blocking solution) in mouse monoclonal anti-GFAP (cat. # 53-9892-82; Thermo Scientific). After washing in PBS (15 min × 3), sections were mounted in ProLong^®^ Gold Antifade reagent (cat. #8961S, Cell Signaling Technology, New York, NY, USA) containing DAPI and cover-slipped. Images were acquired with a Leica TCS SP8X confocal system (Leica Microsystems, Mannheim, Germany) using an HCX PL APO 40_/1.30 Oil objective or for low magnification HCX PLS-APO 5× and appropriate excitation with 405 and 488 nm lasers, which were then analyzed with LAS AF software (Version 2.6 Light, Leica Microsystems, Mannheim, Germany) and ImageJ 1.47 software (NIH, Bethesda, MD, USA). The brightness and contrast of images were adjusted in a standardized manner for all images as previously described [32,33]. All images and graphs were generated and assembled in figures using SPSS (version 6) and Microsoft PowerPoint (Microsoft Office 10). Colocalization between prominin-1 or ICAM-1 and GFAP was determined using the ImageJ plugins (Just Another Colocalization Finder (JACoP) and Colocalization Finder (NIH, Bethesda, MD, USA)). Pearson’s (r) and Mander’s (M) correlation coefficients were used as determinants of red and green fluorescence signal alignment. Complete colocalization was considered equal to 1.

### 2.4. Fluorescence In Situ Hybridization and Confocal Microscopic Analysis of lncRNAs PARTICLE and GAS5

Procedures were carried out as previously reported [35,36] and in accordance with Stellaris fluorescence in situ hybridization (FISH) instructions (www.biocat.com (accessed on 20 July 2022)). Through an online probe design tool (www.biosearchtech.com/stellarisdesigner/ (accessed on 10 August 2022), specific probes were chosen from input sequences (*PARTICLE* NR_038942.1; *GAS5* NR_152521.1) for optimal binding properties to the target RNA. Probe fluorophores 5′carboxyfluorescein FAM (excitation (Ex): 495 nm; emission (Em): 520 nm) were chosen for the detection of these lncRNAs. Confocal fluorescence microscopic imaging was acquired with a Leica TCS SP8X confocal system (Leica Microsystems, Mannheim, Germany) using an HCX PL APO 40_/1.30 Oil objective and appropriate excitation using 488 nm and emission using 509 nm lasers. Emitted fluorescence signals were sampled at a resolution of 30 nm/pixel with a dwell time of 1.5 μs. Image analysis was carried out as indicated above.

### 2.5. RNA Isolation and Real-Time Quantitative PCR from GBM Pathology Sections

A modified protocol for long non-coding transcript isolation from paraffin embedded sections was utilized [37]. In brief, this involved GBM removal from microscope slides with a blade cleaned with xylene followed by deparaffinization. Protein digestion was carried out with lysis buffer (20 mM Tris HCl pH 8.0, 1mM CaCl_2_, 0.5% sodium dodecyl sulphate, 5 mg proteinase K (cat. # 19131, Qiagen, Venlo, The Netherlands)) and with incubation at 56 °C for 1 h and immediate transfer to ice. Trizol reagent (cat. # T9424, Sigma Aldrich, St. Louis, MO, USA) with chloroform (cat. # 17110, Penta Chemicals, Prague, Czech Republic) extraction was added, followed by centrifugation for phase separation. Isopropanol (cat. # 17500-11000, Penta Chemicals) was added to the upper layer with RNA, followed by O/N precipitation. After centrifugation, RNA pellets were air dried and resuspended in nuclease free water with purity assessment using O.D. 260/280 ratio determination (NanoDrop 1000, Thermo Scientific). RNA was treated with DNase I for 20 min at 37 °C to remove genomic DNA. Total RNA (1 μg) was converted into first strand cDNA using standard protocol procedures (with the inclusion of random hexamers) and reagents from Life Technologies, Darmstadt, Germany. Quantitative PCR for the determination of lncRNA *PARTICLE* utilizing the following reaction conditions: cDNA (50–100 ng), 1 × Taqman universal PCR master mix (no AmpErase UNG; Life Technologies, cat. # 4324018) and 1 × Taqman *PARTICLE* expression assay (cat. # Hs03847241_s1, Thermo Scientific).

### 2.6. Statistical Analysis

A two tailed Student’s *t*-test was employed for comparative purposes between samples (Excel, Microsoft Office, Version 10). Data are presented as mean ± standard error with alpha significance determined at *p* = 0.05. Interquartile range represented the median and data range distribution (SPSS version 28 software, IBM). Experiments were replicated three times for patients (*n* = 4, total R.O.I = 10 per sample) and Eker rats (cortical sections obtained from a previous study involving *n* = 6 [32], R.O.I = 10 per sample).

## 3. Results

### 3.1. Morphological Correlates of Malignant Glioma/GBM in Patient Histopathology

Individuals represented responders or non-responders to temozolomide chemotherapy (Table 1), with the hallmarks of GBM (according to WHO diagnostic criteria [16]) evident in all patients. While the tumor margins were poorly defined, heterogeneous-shaped clusters of tumor cells could be seen intercalating with necrotic regions (Figure 1A) and (Appendix A). The tumors were highly vascularized (Figure 1B), with proliferative cells preferentially adjacent to newly formed microvessels (Figure 1C). Tumor cells, with heterochromic polymorphic nuclei, were associated with regions showing extensive necrosis (Figure 1D). Complex anatomical features included zones of proliferating cells coinciding with areas of complete cellular depletion, which were caused presumably by apoptotic events (Figure 1E). The tumors were of high cellularity, with an apparent extensive mitotic activity that resulted in pleomorphic nuclei (Figure 1F). This tended to cause a high nuclear-to-cytoplasmic ratio in tumor cells (Figure 1F). Such features of malignant GBM confirmed the clinicopathological characteristics in these classical GBM patients.

### 3.2. Prominin-1 Independent from Astrocytes Elicits Elevated Expression in Pseudo-Palisades Adjacent to Necrotic Foci

The astrocytoma basis of GBM prompted the examination of prominin-1 expression in astrocytes in patient samples. Analysis revealed a lack of colocalization of prominin-1 in this cell type in pseudo-palisades or surrounding tissue (Pearson’s correlation coefficient r = 0.386 or r = 0.482, respectively; Figure 2(A1–A6)). Equal intensity threshold values for green or red fluorescence micrographs showed divergent Mander’s colocalization coefficient values in pseudo-palisades versus the surrounding tissue (Figure 2(A6)). It has been proposed that pseudo-palisades are hypoxic tumor cells actively migrating away from a central vascular lumen which has either degenerated or thrombosed [38]. This unique feature of malignant glioma was evident in all four GBM patient pathology sections, as well as in the Eker rat cortical brain regions examined in this report. All human GBM slides contained expansive necrotic regions (Figure 2B) within which were ovoid elongated structures predominated by prominin-1 expression along their peripheral palisades (Figure 2(B1–B5)). A similar pattern was evident in the Eker rat cortex, whereby prominin-1 showed a specific association with ring-like structures within degraded tissue (Figure 2(C1–C3)). Previously shown to be widely distributed within U87MG [26], dispersion data indicated that the median prominin-1 signal was higher in the Eker rat cortex (median = 110.6 arbitrary units (a.u.)) than in the GBM patients (46.5–92.33 a.u.) (Figure 2D). The prominin-1 signal intensity range for patient 3 (P3) and patient 4 (P4) were identical at 50.5 a.u. for both cases, though P3 had a lower interquartile range (IQR: 27.15 a. u.) compared to P4 (IQR: 45.05). Similar mean prominin-1 intensity values were also determined for these individuals (P3: 52.17 ± 18 a.u.; P4: 58.85 ± 21 a.u.; Student’s *t*-test *p* = 0.784). Both patients had undergone similar chemotherapeutic adjuvant therapy (Table 1), potentially contributing to the alignment of prominin-1 signal intensity ranges within their respective pathological specimens. The IQR for patient 1 (P1) was lower and was positively skewed, while the IQR for patient 2 (P2) was wider and had a negative skew. The dispersion range for prominin-1 in the Eker rat cortex was short (29.66 a.u.; IQR = 19.65 a.u.) and almost symmetrical. When the focus is placed solely on the ovoid pseudo-palisade areas, significant differences emerged with prominin-1 expression between humans and rodents. The percentage of prominin-1 signal per pseudo-palisade area (R.O.I = 10) was significantly higher in the Eker rat cortexes compared to all GBM patients (P1, p = 0.0028; P3, *p* = 0.0005; P4, *p* = 0.0028; Student *t*-test), except for one individual (P2, *p* = 0.55) who had been intolerant to chemotherapeutic adjuvant intervention (Figure 2E). While prominin-1 appears confined to similar areas in rodent and human GBM sections, variable expression was noted between species. Given the varied prominin-1 expression between GBM cases, future larger studies are envisioned to determine whether this is reflective of previous therapeutic interventions. Future perspectives should consider proteomic investigations of prominin-1 containing membrane particles (not exosomes) in human body fluids (e.g., human urine, seminal fluid and saliva) to diagnose diseases involving the down-regulation of stem-cell properties or differentiation [39].

### 3.3. Chemotherapeutic Target ICAM-1 Associates with Degenerating GBM Vasculature

Our previous study noted incremental ICAM-1 in response to increasing ATRA dosage [26]. It has been suggested that targeting this molecule may provide a strategy for enhanced efficacy of anti-angiogenic GBM therapy [26]. While ICAM-1 appeared widely distributed in human GBM and Eker rat cortical sections, patterns of its specific expression emerged especially in human pathological samples. Of note, ICAM-1 did not predominate in astrocytes (Pearson’s correlation coefficient r = 0.327, Figure 3A,E). High ICAM-1 intensity was noted in small ovoid structures, perhaps representing zones of microvascular hyperplasia (Figure 3B,F; zone = 1). Whether such regions signified the development of new blood vessels could be speculated given the proximity to necrotic pseudo-palisading tissue within which tumor cells potentially migrated towards the emerging vasculature (Figure 3B,F; zone = 2). Of note, low levels of ICAM-1 expression were demonstrated throughout human GBM (Figure 3B,F; zone = 3). ICAM-1 was also intensely expressed within centralized tumor cells, as well as in the surrounding pseudo-palisade circular boundary (Figure 3C). Eker rat cortical tissue revealed significantly higher ICAM-1 signal per area compared to all human GBM patients examined (Figure 3D,G), along with a generalized distribution pattern that tended to align with blood vessels (Figure 3D asterisk). This was reflected in upper-end values within the IQR for ICAM-1 in the Eker rats (median = 107 a.u., min. − max. = 96–129 a. u.) compared to GBM patients (Figure 3H). Significant differences were noted within GBM patients when comparing the percentage of ICAM-1 expression per area (R.O.I = 10). While similar levels were noted for P3 and P4 (*p* = 0.06), differences in ICAM-1 were found when P2 was compared to P3 (*p* = 0.0005) or P4 (*p* = 0.0037). This was supported by overlapping dispersion ranges for P3 (median = 50.87 a.u., min. − max. = 47–60 a. u.) and P4 (median = 63.94 a.u., min. − max. = 50–71 a. u.) (Figure 3H). The IQR of P1 also overlaps with these two patients, while P2 showed a negative skew towards the higher intensity ICAM-1 range (Figure 3H). As the chemotherapeutic intervention for P1 was unknown, conclusions cannot be drawn on its influence on ICAM-1 expression. Elevated ICAM-1 was found in the patient (P2) with chemotherapeutic non-tolerance. Nevertheless, ICAM-1 levels in P1 align with both responders to adjuvant therapy, which tends to suggest a potential therapeutic influence on this biomolecule. This will be deciphered in a future larger experimental cohort.

### 3.4. Long Non-Coding RNA PARTICLE Expression in Tumor Cells within Necrotic GBM Degenerative Zones

*PARTICLE* has been recognized as a silencer of tumor suppressor genes and a maintainer of cancer viability [35,36,40]. Our previous study demonstrated the significant curtailment of *PARTICLE* by ATRA at the leading edge of U87MG [26]. This contrasted with its sustained elevated expression with resistance to ATRA treatment within the established cellular monolayer microenvironment [26]. Likewise, this study reveals the high nuclear expression of *PARTICLE* in tumor cells still surviving in necrotic zones in human GBM. In situ hybridization confocal micrographs of *PARTICLE* in P1–P4 GBM pathological sections were converted to optical density threshold images. Maximum intensity levels for *PARTICLE* were associated with the upper end threshold levels for tumor clearance zones (Figure 4A). Despite being expressed in both nucleus and cytoplasm in other cancer types (e.g., breast and osteosarcoma; [35,36]), *PARTICLE* expression is confined to the nucleus in human GBM (Figure 4B,C). Intriguingly, *PARTICLE* tended to predominate along one side of a necrotic clearance zone (Figure 4D) and to be also expressed in atypically shaped nuclei (Figure 4E). A highly significant 1.93-fold increased difference (65.9%, *p* = 0.008) in *PARTICLE* was noted between such peripheral regions (Figure 4D,F, demarcated zones I and II). Quantitative real-time PCR detection successfully amplified *PARTICLE* from deparaffinized pathology sections, highlighting the robust elevated expression of this non-coding transcript in GBM. Given the role of *PARTICLE* as a tumor activator in several cancers examined to date [35,36], it can be speculated that this lncRNA assists in GBM survival in proximity to zones of degeneration.

### 3.5. GBM Necrosis Associated with Elevated Long Non-Coding RNA GAS5 in Chemotherapeutic Responsive Patients

It has been reported that elevated *GAS5* is closely associated with cancer patients overall survival [30,41,42,43] and acts as a functional tumor suppressor in the U87MG cell line [44]. This investigation found *GAS5* in necrotic regions of human GBM pathological sections. Lower levels of *GAS5* were detected in proliferative regions in GBM compared to centralized degenerative zones (Figure 5A). In situ hybridization confocal micrographs of *GAS5* expression in P2–P4 GBM pathological sections were transformed to optical density threshold images. Maximum intensity levels for *GAS5* were associated with upper-end threshold levels for deep tumor necrotic zones (Figure 5B). GBM sections derived from the patient who was non-tolerant to adjuvant therapy (P2) showed preferential *GAS5* expression in these hypoxic regions compared to its expression in more structurally intact tumor cells (39% increase; P2; Figure 5A,B). Further examination of patients tolerant to chemotherapy (P3, P4; Figure 5C), demonstrated a similar *GAS5* distribution pattern to GBM sections from P2 (Figure 5A,B). Nevertheless, a significantly greater mean *GAS5* intensity was revealed in necrotic regions compared to adjacent proliferative zones (Figure 5C,E, 82% increase; *p* = 0.003). Of note, *GAS5* expression was perinuclear-cytoplasmic in the GBM intracellular distribution (Figure 5D). While *GAS5* could be detected with in situ hybridization, amplification of *GAS5* from RNA extracted from deparaffinized GBM sections proved unsuccessful. This investigation has examined GBM sections from patients with alternative responses to adjuvant therapy. While *GAS5* predominated in degenerating tumor cells in all patients, it can be speculated that patients who responded to chemotherapeutic intervention demonstrated even further elevated levels of this tumor suppressor than non-responders, perhaps influencing an extension in overall survival.

In summary, prominin-1, ICAM-1 and two long non-coding RNAs *PARTICLE* and *GAS5* elicited specific territorial patterns of expression within the GBM pathological sections. Significant signal intensity differences between GBM patients open further avenues worth exploring in the quest for effective targeted treatment strategies.

## 4. Discussion

The nuances associated with evaluating U87MG proliferation have been previously conveyed, enabling this glioblastoma cell line model to be suitable for stem cell differentiation and chemotherapeutic resistance studies [26,45,46,47]. This conclusion was reached from cell-based experimentation focused on the response of the stem cell marker prominin-1, the chemotherapeutic target ICAM-1 and the alternative expression profiles of the lncRNAs *PARTICLE* and *GAS5* within this in vitro GBM microenvironment [26]. Herein, we present the pre-clinical applicability of these important biomolecules through analytical expansion to a representative glioma animal model and GBM patients that elicited alternative responses to adjuvant chemotherapy.

An in-depth microscopic evaluation, identifying necrotic foci, enhancement of cellular proliferation and vascular remodeling, concluded that the Eker rat was a suitable model for primary brain tumor studies [32,48]. This investigation recognized such morphological correlations of malignancy in GBM patient pathological sections from haemoxylin and eosin staining. Immunofluorescence imaging highlighted an elevated prominin-1 expression in pseudo-palisades adjacent to necrotic foci in both animal and human GBM. By extending our previous work [26], we revealed ICAM-1 to be associated with degenerating vasculature within patient GBM and rodent subcortical tumors. In situ hybridization experiments demonstrated that the lncRNA *PARTICLE* elicited a predominantly nuclear intracellular expression profile potentially activating tumor survival within necrotic degenerative GBM zones. Such regions of tumor necrosis were associated with elevated lncRNA *GAS5* expression in chemotherapeutic responsive patients, supporting its role as a tumor suppressor and prognostic biomolecule of extended overall survival.

This investigation centered on the evaluation of GBM pathological sections obtained from patients who succumbed within twelve months of diagnosis. Ranging from 57 to 77 years, this group included female and males of unknown and mixed responsiveness, respectively, to temozolomide treatment. Histopathological analysis of brain tumor sections from these patients identified typical GBM characteristics [49,50]. These features included atypical cells with hyperchromic polymorphic nuclei [49] and the presence of necrotic regions coinciding with high cellular zones, potentially representative of tumor cell migration from degrading vasculature destined for further microvessel formation [51]. The relevance of such pseudo-palisades around necrotic vessels is believed to be an underappreciated feature that contributes to GBM clinical progression [52]. Reports have recognized the perivascular ‘railway-like’ expression profile of the stem cell marker prominin-1 (CD133+) within GBM [53]. Such striped patterns of prominin-1 expression were noted in our samples on occasion and strongly support the predominant presence of prominin-1 within the peripheral zones of ovoid structures, representing blood vessels within human GBM and Eker rat cortical sections. Importantly, our findings align with the first identification of prominin-1 expression within tumor cells of pseudo-palisade formations that delineate necrosis within GBM [53]. This study noted higher prominin-1 expression in the rodent samples compared to GBM patients who responded to TMZ treatment. Given similar elevated prominin-1 signals between Eker rats and a TMZ non-tolerant patient compared to treatment-responsive patients, it can be speculated that prominin-1 may be worth further investigation in this regard using a larger cohort. This is supported by previous studies calling for further investigation of intercellular communication between prominin-1 (CD133^+^) niches and adjacent blood vessels within GBM, with the view to design specific targeted treatments directed at brain tumor stem cells [53]. Of note, CD133^+^ cells elicited greater tumorigenic potential than CD133^-^ cells in response to ionizing radiation [54]. It has been suggested that CD133+ tumor cells confer glioma radio-resistance and act as a potential source of tumor recurrence after radiation [55]. The basis for therapeutic resistance has yet to be fully deciphered, yet evidence of the involvement of microvesicle-mediated transfer of the metabolic enzyme nicotinamide phosphoribosyltransferase (NAMPT) to radiosensitive cells has been reported [56].

GBM is considered one of the most angiogenic tumors characterized by microvascular proliferation involving tumor cell transformation in a process of tubular vasculogenic mimicry [57,58]. In contrast with reports of the association of strong vascular ICAM-1 staining in GBM zones with fewer blood vessels [58], this study identified elevated ICAM-1 expression in ovoid structures most likely representing blood microvessels. ICAM-1 has been reported to be rapidly induced after the onset of severe hypoxia [59], supporting the findings from this study whereby ICAM-1 elicited intensive expression within centralized tumor cells bounded by circular pseudo-palisades. While the implication of increased ICAM-1 in GBM has not been fully elucidated, the results presented herein point to significant differences in its expression between patients with varying adjuvant therapy tolerance. Nevertheless, in larger studies staining intensity of vascular ICAM-1 was not associated with GBM patient survival [58].

Recognition of the functional importance of lncRNA in the context of GBM has recently emerged following their identification as master regulators of tumorigenesis [60,61,62]. Nevertheless, limited mechanistic studies of lncRNAs exist in relation to their role in GBM etiology, with most studies executed using traditional serum-growth cell lines, such as U87MG [26,61]. The lncRNA *PARTICLE* is a key epigenetic mediator involved in the intercellular communication and recruitment triplex platform for down regulators of tumor suppressor genes in response to irradiation [35,36]. The in vitro investigation of *PARTICLE* demonstrated its robust expression in non-irradiated U87MG and its curtailment in response to ATRA [26]. The view that lncRNA expression is dynamic across multiple single cells in GBM tumors and cell lines [60] was supported by *PARTICLE* being responsive to ATRA only within the U87MG leading edge cells, yet elicited resistance to this active metabolite of vitamin A in the established monolayer [26]. This investigation extended the assessment of *PARTICLE* to human GBM pathology and demonstrated its significantly elevated expression in tumor cells adjacent to necrotic GBM degenerative zones. The role of *PARTICLE* as an enhancer of cancer viability has been demonstrated [36]. The predominance of *PARTICLE* expression along one side of a necrotic tumor niche is perhaps indicative of its activation potential for tumor cell survival and proliferation as a regulator of the dominant growth direction in GBM.

In contrast, the tumor suppressor *GAS5* has been reported to be decreased in GBM and associated with the overall and disease-free survival relative to patients with high expression of this lncRNA [63,64]. This study demonstrated that *GAS5* predominated in degenerating GBM extracted from patients with alternative responses to adjuvant therapy. While a similar *GAS5* distribution pattern was apparent in all GBM patients, those that responded to temozolomide revealed significantly higher levels of this tumor suppressor.

The limitations of this investigation center on the patient cohort size and findings are likely to be confirmed in a larger-scale study. Emphasis was placed solely on biomolecules previously studied in vitro using a representative GBM cell line. Eker rat data were confined to prominin-1 and ICAM-1 investigations due to potential conservation issues with lncRNA assessment. The merits of this study center on encouraging in vitro research to be extended into the assessment of patient-derived tissue. Our findings for prominin-1, ICAM-1 and *GAS5* in GBM support the data of other research groups [53,59,61,65]. In addition, this investigation is the first to show the predominant expression of *PARTICLE* in GBM pathology, facilitating future research into this important tumor activator. The technical aspects of clinical implementation involve challenges associated with simultaneous examination of prominin-1, ICAM-1 immunohistochemistry combined with *GAS-5* and *PARTICLE* in-situ hybridization. In addition, overcoming the blood-brain barrier needs to be considered for effective GBM patient treatment.

## 5. Conclusions

This study highlighted the specific expression pattern of prominin-1, ICAM-1, *PARTICLE* and *GAS5* within GBM histological sections. While cell lines provide a platform for biomolecule response to controlled media, extending investigations into the complex environment of GBM sections makes us a step closer to future promising biomarkers for GBM detection, therapeutic response and prognosis prediction.

## Figures and Tables

**Figure 1 pharmaceutics-16-00123-f001:**
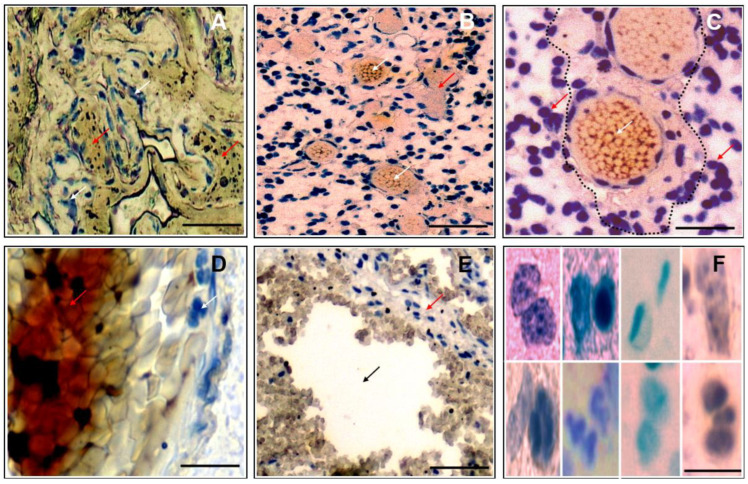
Histopathological features of human glioblastoma multiforme (GBM) following surgical resection and hematoxylin and eosin staining. Representative light micrographs showing (**A**) heterogeneous shaped clusters of tumor cells (white arrows) intercalating within necrotic regions (red arrows). Scale bar 200 μm. (**B**,**C**) Highly vascularised tumor regions with proliferative cells (red arrows) adjacent to microvessels (white arrows). Scale bar 200 μm (**B**) and 100 μm (**C**), respectively. (**D**) Tumor with heterochromic polymorphic nuclei (white arrow) associated with extensive necrosis (red arrow). Scale bar 100 μm. (**E**) Proliferating cells (red arrow) coinciding with apoptotic zones (black arrow). Scale bar 200 μm (**F**) Representative examples of tumor cells with pleomorphic nuclei. Scale bar 50 μm.

**Figure 2 pharmaceutics-16-00123-f002:**
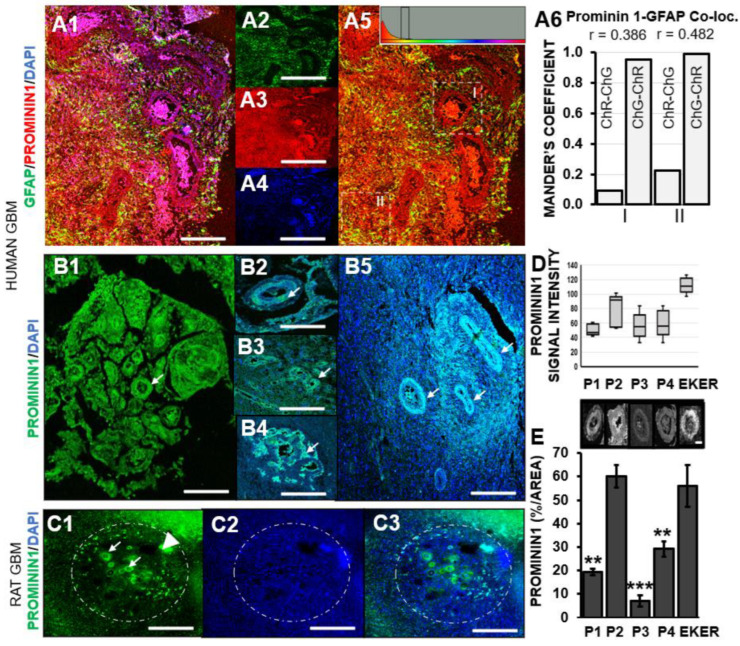
Prominin-1 does not colocalize with the astrocyte marker GFAP but is expressed in circular zones of palisading endothelial cells in GBM. (**A1**) Representative confocal micrograph of merged images for GFAP (astrocyte marker) and prominin-1 in human GBM. Scale bar 250 μm. (**A2**–**A4**) Separated fluorescence images representing GFAP (green; **A2**), prominin-1 (red; **A3**) and nuclei stained with DAPI (blue; (**A4**)). (**A5**) Cytofluorogram with threshold indicator in yellow zone. (**A6**) Histograms of Mander’s coefficient of colocalization for zones I and II (as shown in (**A5**)). (**B**,**C**) Representative confocal micrographs showing prominin-1 expression in human GBM and Eker rat cortical brain sections. (**B**) Immunohistochemical detection revealed elevated prominin-1 expression in palisading endothelial cells located in circular zones surrounding necrotic regions in human GBM pathological sections (white arrows). Scale bar 2 mm (**B1**), 1 mm (**B2**–**B5**). (**C**) Prominin-l expression in rodent GBM model ie. Eker rats in similar circular zones surrounding necrosis in the brain. Scale bar 2 mm. **C1**—Prominin; **C2**—Dapi, **C3**—merged image) (**D**) Interquartile range of prominin-1 signal intensity in pathological GBM sections (ROI = 10 per patient (P) or EKER rat). (**E**) Representative grey scale micrographs showing prominin-1 in circular cellular zones within GBM above histogram quantitation of percentage expression per area. Asterisks represent significant differences in the levels of prominin-1 in Eker rats compared to patients (P1, P3 and P4), ** *p* ≤ 0.005; *** *p* ≤ 0.0005.

**Figure 3 pharmaceutics-16-00123-f003:**
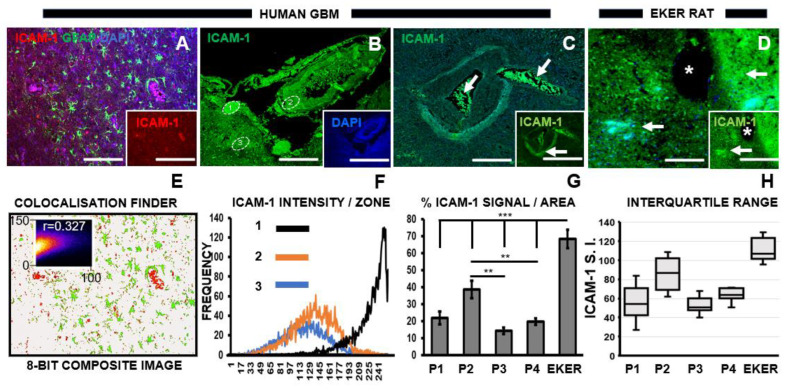
ICAM-1 does not colocalize with the astrocyte marker GFAP but reveals predominance adjacent to zones of tumor decay. (**A**) Representative confocal micrograph of merged images for GFAP (astrocyte marker) and ICAM-1 (red; inset) in human GBM. Scale bars 250 μm. (**B**–**D**) Representative confocal micrographs showing ICAM-1 expression in human GBM and Eker rat cortical brain sections. (**B**,**C**) ICAM-1 (green) signal detected in human GBM. Numbered ROI were analyzed for ICAM-1 signal intensity as shown in panel F, showing the high signals (zone 1) close to regions devoid of cellular content. Inset indicates dual immunofluorescence staining for both ICAM-1 (green) and nuclei (DAPI, blue). (**C**) High levels of ICAM-1 are located within (arrow) and surrounding circular zones (arrow, inset) in GBM sections. (**D**) ICAM-1 (green) reveals a highly distributed expression pattern (arrow) in EKER rat model compared to human GBM adjacent to blood vessel (asterisk). Scale bars in main micrographs and insets 2 μm and 1 μm, respectively. (**E**) Composite image (8-bit) showing the lack of colocalization between GFAP and ICAM-1 with scatterplot (inset). (**F**) Frequency polygons of ICAM-1 intensity per zone (1–3 indicated in panel B). (**G**) Histogram showing interpatient comparison of percentage ICAM-1 signal/area and between EKER rats and human GBM patients (P). Asterisks indicate significant differences, ** *p* ≤ 0.005; *** *p* ≤ 0.0005. (**H**) Interquartile range of ICAM-1 signal intensity between GBM patients (P1–P4) and an EKER rat glioma model.

**Figure 4 pharmaceutics-16-00123-f004:**
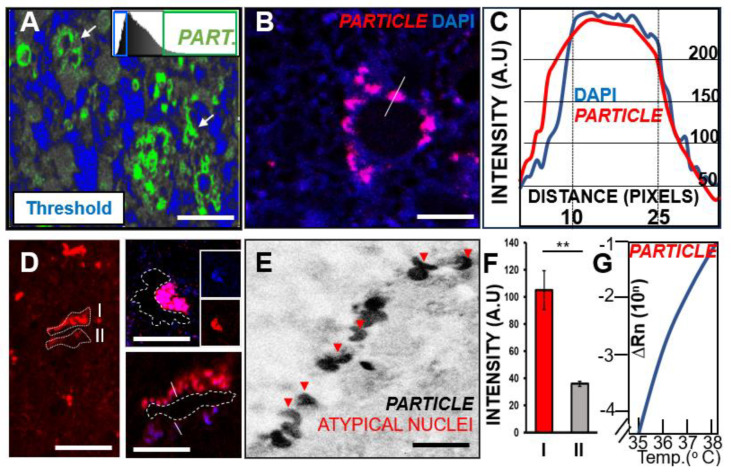
Tumor activator long non-coding RNA *PARTICLE* reveals nuclear expression pattern predominantly in tumor cells adjacent to zones of decay in human GBM. (**A**) Pseudo-colored high threshold image highlighting elevated *PARTICLE* (green) expression in regions adjacent to cellular free zones (blue) within a human GBM pathological section (arrows). Threshold demarcation schematic (upper right). Scale bar 100 μm. (**B**) Representative confocal micrograph of *PARTICLE* expression next to cell free regions of the tumor. Nuclear presence shown with DAPI staining (Blue). Line indicates transection point analyzed in panel C. Scale bar 200 μm. (**C**) *PARTICLE* (red) reveals nuclear (DAPI, blue) expression pattern as shown by representative line graphs of signal intensity. (**D**) Representative confocal low and high magnification micrographs show increased *PARTICLE* (red) expression predominantly along one side adjacent to tumor free zones. Analyzed regions (I, II) demarcated in dashed lines in main micrograph. Scale bar 1 mm. Insets show tumor free zones (dashed lines). Nuclei stained with DAPI (blue, insets). Scale bar 200 μm. (**E**) Grey scale micrograph showing *PARTICLE* expression in atypical nuclei in human GBM (red arrowheads). Scale bar 100 μm. (**F**) Histogram of significant *PARTICLE* intensity signal differences between demarcated zones (shown in panel D) with the GBM tumor. (**G**) Real-time quantitative PCR amplification curve of *PARTICLE* expression in human GBM. ** *p* < 0.005.

**Figure 5 pharmaceutics-16-00123-f005:**
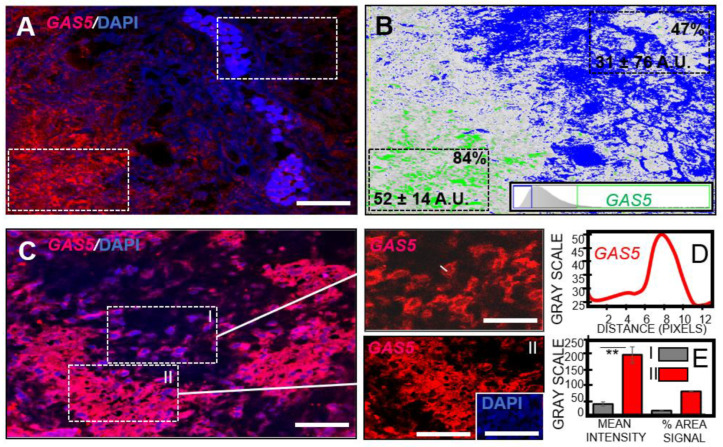
Tumor suppressor long non-coding RNA *GAS5* enhanced in necrotic human GBM. (**A**) Representative confocal micrograph showing *GAS5* (red) and DAPI stained nuclei (blue). Dashed lines indicate regions quantified in panel B. Scale bar 1 mm. (**B**) Pseudo-colored high threshold image highlighting elevated *GAS5* (green) expression in necrotic regions compared to cellular zones (blue-white) within a human GBM pathological section. Threshold demarcation schematic (lower right). *GAS5* intensity and percentage signal per demarcated area (dashed line) indicated. (**C**) Representative confocal micrographs of *GAS5* (red) and DAPI stained nuclei (blue). Rectangle boxes (I, II) further highlighting increased levels of *GAS5* in region II with amorphic nuclei (inset) indicative of tissue necrosis and tumor suppression. Scale bar 50 μm. (**D**) Grey scale line graph profiling *GAS5* in perinuclear/cytoplasmic signal (line shown in panel C upper right). (**E**) Histograms of *GAS5* mean signal intensity and percentage area of expression in regions I and II (panel C). Asterisk represent significant differences, ** *p* ≤ 0.005.

**Table 1 pharmaceutics-16-00123-t001:** Clinicopathological characteristics of patients involved in this study who were diagnosed with glioblastoma multiforme (WHO Grade IV) and treated with surgical resection with/without temozolomide (TMZ). Abbreviations: NT: temozolomide not tolerated; NK: temozolomide treatment not known; Met.: metastasis; IDH: isocitrate dehydrogenase 1; WT: wild-type; +: presence of the *IDH* R132H mutation.

Patient Number	Identification Number	Gender	Birth Year	Diagnosis Year	Death Year	Primary Tumour	IDH	TMZ	Met.
P1	5488/19	Female	1962	2019	2019	Yes	WT	NK	No
P2	11724/19	Male	1949	2019	2020	Yes	+	NT	No
P3	47/19	Male	1952	2019	2020	Yes	WT	Yes	No
P4	286/19	Male	1943	2019	2020	Yes	WT	Yes	No

## Data Availability

The data presented in this study are available in this article and Appendix A).

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
