# Peer review of "Biomolecules to Biomarkers? U87MG Marker Evaluation on the Path towards Glioblastoma Multiforme Pathogenesis"

_pharmaceutics, 2024, doi:10.3390/pharmaceutics16010123_

Round 1

Reviewer 1 Report

Comments and Suggestions for Authors

The authors submitted a research article in which they evaluated  U87MG proliferation in connection with the presence of lnc-RNAs and morphological varians of glyoblastoma. The authors found the specific expression pattern of prominin-1, ICAM-1, PAR-TICLE and GAS5 within GBM histological sections. Yet, they concluded that these findings may be new platform fot the future investigation with the aim of identifying new promising biomarkers for
GBM detection, therapeutic response, and prognosis prediction. The aim of the study is clear. The structure of the paper is logic and the paper contains well-balanced subsections which cover all aspects of the initial hypothesis. The figures and tables are legible. The benefit of the study is thorough combination of the biomarker' expression with morphological variants of GBM. The weaknes of the study is a smal sample size. However, the findings are intriguing and the methodology is elegant. I would like to congratulate the authors on the study. However, there are some aspects which require to be discussed.

1. The authors did not report their opinion regarding investigatiopn of the proteomics od extracellular vesicles as future perspectives. Please, extend appropariate subsection interpretating the aspect mentioned above.

2. Please, add at the and of the section "Discussion" clear explanation regarding technical aspects of the clinical implementation

Reviewer 2 Report

Comments and Suggestions for Authors

In the presented article entitled: Biomolecules to Biomarkers? U87MG marker evaluation on the path towards Glioblastoma Multiforme Pathogenesis authors have taken into consideration the GMB biomarker to detection, therapeutic response and therapy promising for brain cancer. Cancer of CNS is a huge and significant problem for both chemo and radiotherapy. Therefore, the objective of the article is very well situated in pharmacology. The subject of the proposed at the manuscript therapeutic problem is derived from the nature of the blood-brain barrier. From the medical point of view, the Glioblastoma Multiforme become the most dangerous due to the lack of well-targeted and selective medical treatment. Moreover, the high variability of GBM makes it difficult for medicalisation. The last one is the main problem for well-targeted therapy what is parallel with other studies. Therefore, the definition of GBM before treatment is the reasonable preclinical validation which increases the % success of therapy. During the reviewing process, I did not find the points for methodology criticism – the materials and methods part were described in a suitable manner.

However, the diagnostic target should be very clearly mentioned in the abstract of the article. Moreover, the article is well-written and readable. However, I recommend authors make the manuscript more compact and simpler. Additionally, it will be also wondered if authors take the decision to write some part of BBB delivery. The above increases the significance of their studies. Moreover, the figures presented the obtained results of experiments should be improved in the resolution. Finally, the conclusion part has taken together the results described in the manuscript clearly and shortly. The used references were correctly cited.

In conclusion, the article is interesting and can be published after cosmetic correction.
